# New Developments in Celiac Disease Treatment

**DOI:** 10.3390/ijms24020945

**Published:** 2023-01-04

**Authors:** Mariana Verdelho Machado

**Affiliations:** 1Gastroenterology Department, Hospital de Vila Franca de Xira, Estrada Carlos Lima Costa, Nª 2, 2600-009 Vila Franca de Xira, Portugal; mverdelhomachado@gmail.com; Tel.: +351-263-006-500; 2Clínica Universitária de Gastrenterologia, Faculdade de Medicina, Universidade de Lisboa, Avenida Prof. Egas Moniz, 1649-028 Lisbon, Portugal

**Keywords:** celiac disease, gluten-free diet, pharmacological treatments

## Abstract

Celiac disease (CD) is a common autoimmune disease affecting around 1% of the population. It consists of an immune-mediated enteropathy, triggered by gluten exposure in susceptible patients. All patients with CD, irrespective of the presence of symptoms, must endure a lifelong gluten-free diet (GFD). This is not an easy task due to a lack of awareness of the gluten content in foods and the extensive incorporation of gluten in processed foods. Furthermore, a GFD imposes a sense of limitation and might be associated with decreased quality of life in CD patients. This results in gluten contamination in the diet of four out of five celiac patients adhering to a GFD. Furthermore, one in three adult patients will report persistent symptoms and two in three will not achieve full histological recovery when on a GFD. In recent years, there has been extensive research conducted in the quest to find the holy grail of pharmacological treatment for CD. This review will present a concise description of the current rationale and main clinical trials related to CD drug therapy.

## 1. Introduction

Celiac disease (CD) is an autoimmune disease in which, in susceptible subjects, the ingestion of gluten triggers an immune attack on the small bowel, as well as a serological response [1]. Unlike other autoimmune diseases, the immunogenic antigens that trigger the immune response in CD have already been identified and highly characterized [2]. As such, removing those antigens by enduring a gluten-free diet (GFD) is a known effective treatment for CD.

The prevalence of CD is estimated to be 1% of the population [3]. Considering a global population of eight billion people, there are roughly eighty million patients with CD. The classical presentation of CD is malabsorption manifestations such as diarrhea, weight loss, and nutrition deficits, although most patients will remain asymptomatic or with non-specific and extraintestinal symptoms [4]. Even though not consistent among all cohorts [5], patients with CD seem to present increased mortality [6,7], which might be mitigated by achieving mucosal healing through a GFD [8,9].

In the last decade, intense efforts have been applied in the search for pharmacological treatments for CD. The bar is high as dietary treatment is already available, which is effective and has no predictable adverse effects. However, adhering to a strict GFD is challenging, resulting in self-reported adherence rates ranging from 42% to 91% [10]. Furthermore, up to 80% of GFD-adherent patients might have inadvertent gluten contamination in their diet [11]. Lastly, mucosal healing after a GFD seems to occur in less than half of adults with CD [12]. Up to 0.5% of patients with CD will progress to refractory celiac disease (RCD) [13], which when associated with aberrant monoclonal T cell infiltration (RCD type 2) presents a high risk of progression to enteropathy-associated T cell lymphoma (EATL) [14].

Drug discovery for treating CD, as a complement to a GFD, aims to intervene in different scenarios, including (a) maintenance therapy, (b) rescue therapy after acute gluten exposure, and (c) the mitigation of inadvertent chronic gluten exposure [15].

This review will summarize the rationale that has been applied to CD drug development, as well as the clinical advances in its research.

## 2. Gluten-Free Diet: The Current Standard of Care

Sensum strictum, gluten refers to proteins present in wheat that contain two major protein components: monomeric water-soluble gliadins and multimeric water-insoluble glutenins. The proteins secalin, in rye, and hordein, in barley, share similar immunogenic properties and have also been called gluten in sensum latum [16]. Oats are phylogenetically more distant and are usually tolerated by CD patients [17].

Patients with CD independent of the presence of symptoms, symptomatic potential CD (that is patients with positive anti-transglutaminase antibody and normal duodenal histology), dermatitis herpetiformis, and gluten ataxia should follow a lifelong GFD, since a GFD is currently the only proven treatment for CD [18]. Regarding patients with asymptomatic potential CD, only a minority will develop villous atrophy and should be followed rather than be recommended a GFD [19].

CD is associated with a panoply of extraintestinal manifestations and autoimmune diseases, such as neuropsychiatric [20,21,22] and dermatological manifestations [23], unexplained abnormal liver enzymes [24], type 1 diabetes mellitus, and autoimmune thyroid disorders [25]. Adherence to a GFD may improve most of those manifestations. Regarding neuropsychiatric manifestations, it is known to improve headache [26] and gluten ataxia as long as there is no irreversible loss of Purkinje cells in the cerebellum [27]. A GFD may also improve response to drug-resistant epilepsy [28], gluten-induced cognitive impairment [29], psychiatric disorders (particularly anxiety) [26], and fatigue in up to 50% the patients [30], while response to peripheral neuropathy is variable with lower response rates in patients presenting anti-neuronal antibodies [31]. Dermatitis herpetiformis typically responds to a GFD [32]. Psoriasis [33], chronic urticaria [34], and recurrent aphthous stomatitis may also improve in patients with CD, while dental enamel defects are irreversible [35]. Furthermore, a GFD may improve metabolic control in CD patients with type 1 diabetes mellitus [33], bone density [36], menstrual disturbances and fertility [37], and unexplained transaminitis [38].

A GFD will only be effective if the patient complies. In most cases, less than 50% of CD patients achieve long-term adherence to a GFD [10]. Indeed, dietary regimens are the least appealing medical strategies, being one of the treatment regimens with the lowest rate of adherence (as opposed to medications that have the highest adherence rate) [39]. A GFD is particularly difficult to follow due to the omnipresence of gluten in processed foods, even in unlikely foods (such as yogurt and frozen fish) and products (such as toothpaste and lipstick). Gluten is the Latin word for “glue”, and its viscoelastic properties make it highly appealing to the food industry. The removal of gluten changes the physical properties and durability of foods. To compensate for gluten removal, the food industry tends to deliver unhealthily high fat and sugar content [40] and decrease healthy supplementation, such as the addition of fiber, iron, folate, and zinc [15]. Gluten-free products are more expensive than their gluten-containing counterparts, even though the recent popularity, and hence widespread availability, of gluten-free products has translated into a decrease in price [41]. Dining out may be challenging as it is difficult to control for gluten contamination, which may subsequently lead to social isolation, anxiety, and impaired quality of life [42].

Another barrier to a GFD is inadvertent gluten contamination. Even lifelong GFD-adherent patients struggle to correctly ascertain gluten content from commercial product labels [43]. Indeed, 70–80% of patients adherent to a GFD present gluten contamination in their diet [11,12,44], with an average gluten exposure of around 150 mg/day [45], which is much higher than the considered safe amount of up to 10 mg/day [46] and higher than the 50 mg/day cutoff (equivalent to 1% of a slice of bread) that is known to be able to induce mucosal atrophy [47]. Of note, the gluten content of a typical Western diet is around 15–20 g/day [48].

Finally, up to 30% of patients report persistent symptoms [49], and one to two-thirds of adults will not achieve histological recovery after one year on a strict GFD [50]. Of note, monitoring of anti-actin IgA antibodies may help to predict GFD-induced resolution of intestinal mucosa damage [51].

## 3. Pathophysiology-Driven Strategies to Treat Celiac Disease

### 3.1. Non-Immunogenic Gluten Delivery

Gluten proteins are rich in proline (15% of amino acid composition) and glutamine (35% of amino acid composition), which confers high resistance to the action of human proteases in the intestinal lumen [16] and results in the production of peptides up to 30–40 amino acids in length that are highly immunogenic [52]. One strategy for the treatment of CD, besides removing gluten from the diet, would be to present non-immunogenic variants of gluten. Different natural wheat variants present different T cell immunogenicity [53]. However, all variants, even ancient ones, seem to present toxic potential [54,55]. Another strategy would be to genetically engineer non-immunogenic wheat. This is not an easy task as around 100 genes encode gluten, and hence silencing one gene would not be enough. Additionally, silencing genes responsible for gluten immunogenicity might hamper its viscoelastic properties [56]. One example is the E82 wheat line produced by RNAi technology that blocks relevant gliadin genes [57]. A pilot study with 21 CD patients eating E82 wheat did show decreased interferon-γ (INF-γ) production in peripheral blood mononuclear cells and a very low level of gluten immunogenic peptides (GIP) in stool samples from those patients, suggesting low exposure to immunogenic epitopes [58].

### 3.2. Blocking Immunogenic Gluten Exposure

In order to induce an immune response, gluten needs to overcome the intestinal epithelial barrier and reach the lamina propria, where it will be presented to the immune system by antigen-presenting cells (APCs). Three strategies have been proposed to neutralize gluten after dietary exposure: (a) digesting gluten through the delivery of exogenous peptidases, (b) sequestering gluten in the intestinal lumen, and (c) decreasing epithelial permeability. These strategies intend to mitigate immune responses to chronic inadvertent low levels of gluten exposure as an adjunctive to a GFD (Table 1).

Regarding the first strategy, human proteases are ineffective at degrading proline- and glutamine-rich gluten proteins. As such, gluten degradation could be achieved by the administration of exogenous endopeptidases that would digest, in the stomach, the gluten proteins into small non-immunogenic peptides before they reach the duodenum. Those endopeptidases must fulfill several requisites: (a) be able to degrade all different gluten immunogenic sequences, (b) be stable and active in the low pH gastric environment and escape gastric protease degradation, and (c) not induce adverse effects in the patient [71]. Several microorganisms express prolyl endopeptidases that have been shown in vitro and in vivo to be able to degrade gluten proteins [72], such as *Aspergillus niger*, *Flavobacterium meningosepticum*, *Myxococcus xanthus*, and *Sphingomonas capsulate* [71]. Similarly, glutamine-specific endoproteases can have a complementary action on gluten degradation. One such example is EP-B2, a cysteine endoprotease expressed in germinating barley (*Hordeum vulgare*) seeds [73].

The most studied endoprotease drug for CD is latiglutenase, formally dubbed ALV003. Latiglutenase is an orally administered mixture of two gluten proteases: ALV001 (a modified recombinant version of glutamine endopeptidase EP-B2) and ALV002 (a modified recombinant version of prolyl endopeptidase from *S. capsulata*). These peptidases have complementary peptidase activity in terms of substrate sequence and length [72]. In 2010, a placebo-controlled pilot study of oral latiglutenase administered with large gluten meals (16 g/day) over 3 days in 20 CD patients abrogated gluten immune responses, with decreased INF-γ secretion by gluten-specific T cells in peripheral blood from those patients [59]. A phase 2a dose-ranging trial administered different doses of latiglutenase (from 100mg to 900mg/day, for 12 or 24 weeks) to 494 CD patients presenting persistent moderate to severe symptoms despite being on a strict GFD for at least one year. Globally, latiglutenase, compared to placebo, did not improve symptoms or histology, i.e., intraepithelial lymphocytosis (IEL) or villous height to crypt depth ratio (Vh:Cd) [61]. However, in the subgroup of seropositive patients, the highest dose of latiglutenase (900 mg) was associated with an improvement in abdominal pain and bloating when compared to placebo [62], suggesting a benefit for patients with gluten contamination in their diet. Subsequently, two phase 2 randomized controlled trials (RCT) in 41 and 43 patients with CD subjected to a 2 g/day gluten challenge for 6 weeks showed that the administration of high doses of latiglutenase (at least 900 mg) was able to prevent mucosal deterioration (abrogated Vh:Cd decrease and IEL infiltration) and improve symptoms (at 1200 mg/day) [60,63]. An ongoing phase 2 RCT (NCT 04243551) in patients with CD subjected to periodic gluten exposure for 6 months is estimated to be completed in May 2023.

Another prolyl endopeptidase that is derived from *Aspergillus niger*, AN-PEP, also showed in vitro gluten-degrading activity [74,75]. Two pilot studies involving 12 healthy subjects and 19 gluten-sensitive patients showed AN-PEP to be capable of achieving significant gastric degradation of gluten [76,77]. A short phase 2 RCT involving 12 CD patients subjected to a 7 g/day gluten challenge failed to demonstrate symptomatic advantages over placebo. However, it should be noted that even the group on placebo did not have worsened symptoms due the gluten challenge [64]. An ongoing phase 4 RCT (NCT 04788797) in patients with CD on a strict GFD was estimated to be completed in December 2022.

More recently, TAK-62, a glutenase effective in vitro [78], was shown to be well tolerated and capable of degrading 97% of gluten in gastric aspirates from CD patients after a gluten challenge [65].

The second strategy is to sequester and neutralize gluten proteins in the intestinal lumen, preventing its digestion in immunogenic gluten peptides. Two main therapies have been studied: AGY, an oral egg yolk anti-gliadin polyclonal antibody [66], and BL-7010, a non-absorbable high molecular weight copolymer of hydroxyethyl methacrylate and styrene sulfonate—P(HEMA-co-SS) [79].

AGY was shown to neutralize gluten proteins in mouse models [80] and decrease symptoms, serology, and intestinal permeability (assessed with lactulose:mannitol excretion ratio, LMER) in a phase 1 open-label single-arm trial involving 10 CD patients on a GFD [66]. An ongoing phase 2 RCT on AGY in symptomatic CD patients on a GFD (NCT 03707730) was estimated to be completed in December 2022.

BL-7010 was shown to bind with high affinity to gliadin in vitro [79], to abrogate gluten-induced intestinal injury in vivo in rodent models [79,81], and to decrease tumor necrosis factor-α (TNF-α) secretion ex vivo in mucosal biopsies from CD patients in the presence of partially digested gliadin [81]. A phase 1 RCT in CD patients was completed in 2014, however no data have been published so far (NCT 01990885).

Lastly, intestinal permeability to gluten has been addressed through modulation of tight junctions. Larazotide acetate, formerly dubbed AT1001, is a synthetic octapeptide structurally related to the zonula occludens toxin (ZOT) produced by the bacterium *Vibrio cholera* [82]. Larazotide improves intestinal barrier function by acting as an inhibitor of zonulin by blocking its receptor [83,84]. Gluten induces zonulin secretion after binding to receptor CXCR3 in enterocytes. Zonulin will then bind to the receptor complex epidermal growth factor receptor (EGFR) and protease-activated receptor-2 (PAR2) in enterocytes, initiating a signaling pathway that is Myd88-dependent and which leads to the polymerization of actin microfilaments and subsequent tight junction disassembly, hence increasing intestinal permeability [85]. A phase 1 double-blind RCT compared larazotide at a dosage of 12mg to placebo administered over 3 days in 21 patients with CD who were subjected to one day of a 2.5 g gluten challenge. Larazotide was associated with fewer symptoms and INF-γ immune response, even though it failed to demonstrate improvements in intestinal permeability (assessed by LMER) [67]. Subsequently, three short-term (14 days [84], 6 weeks [69], and 12 weeks [70]) phase 2 RCTs and one meta-analysis [82] enrolling 626 CD patients, treated with larazotide doses that ranged from 1mg to 8mg three times per day, showed improvements in symptoms after a gluten challenge, but not in patients on a strict GFD. While no data on histology were described, those trials did not demonstrate an improvement in terms of LMER, which might be explained by LMER baseline variability [68]. Interestingly, the authors found an inverse dose–response relationship (lower doses presenting better results), which might be explained by an increase in larazotide fragmentation at higher doses (with larazotide fragments being competitively less effective) or by peptide aggregation compromising its function at higher doses [70]. A phase 3 RCT among symptomatic CD patients on a GFD (NCT 03569007) named CeDLara (Celiac Disease Larazotide) planned to enroll 525 patients. However, an interim analysis of half of the initial target enrollment showed disappointing results, leading to discontinuation of the study in June 2022. Strategies aiming to decrease intestinal permeability are hindered by transcellular routes for gluten which allow it to trespass from the lumen into the lamina propria [1].

### 3.3. Transglutaminase Inhibition

Once gluten reaches the lamina propria, it undergoes deamidation by the enzyme tissue transglutaminase-2 (TG-2), which converts glutamine residues in glutamic acid [16]. This is a critical step for increasing gluten immunogenicity by increasing the stability of gluten-HLA DQ2/8 complex [86]. TG-2 also induces degradation of the anti-inflammatory PPAR-γ and promotes transcellular intestinal permeability to gluten [87]. This enzyme is also the autoantigen for classical serology in CD diagnosis [1]. One strategy that aims to decrease gluten immunogenicity is the inhibition of TG-2 activity.

Inhibition of TG-2 has been shown to abrogate gluten-induced immune activation in vitro [88,89] and ex vivo in intestinal biopsies from CD patients [89], and has also been shown to decrease enteropathy in animal models [90]. Of note, mice deficient in TG-2 develop systemic autoimmunity with splenomegaly and glomerulonephritis due to impaired clearance of apoptotic cells during thymus involution, which should prompt further investigation of the possible side effects of TG-2 inhibition [91]. Furthermore, TG has a role in extracellular matrix remodeling and the repair of mucosal damage, which must be taken into account in therapeutic strategies based on TG inhibition [92].

ZED1227 is an oral first-in-class selective inhibitor of TG-2. Phase 1 clinical trials among 100 healthy female and male volunteers (EudraCT 2014-003044-13 and 2015-005283-42), treated with up to 500 mg of ZED1227, proved it to be safe and well tolerated [93]. Recently, a phase 2 proof-of-concept trial tested increasing doses of ZED1227 (10 mg, 50 mg, or 100 mg) for 6 weeks and compared them to placebo in 160 CD patients undergoing a gluten challenge (3 g/day) [94]. ZED1227 abrogated intestinal mucosa injury (blunting the gluten-induced decrease in Vh:Cd and increase in IEL) and improved symptoms and quality of life. It was well tolerated, with the development of a cutaneous rash in only 8% of patients on 100 mg of ZED1227.

### 3.4. Immune Modulation

At the lamina propria, immunogenic deamidated gluten peptides will be exhibited at the surface of APCs, bound to major histocompatibility complex (MHC) class II proteins in a lock–key fashion. The subtype of human leukocyte antigen (HLA)-DQ2/DQ8 is the lock for that key, and carrying those haplotypes is a necessary albeit not sufficient condition for the development of CD [95]. Those APCs will present gluten epitopes to CD4^+^ T cells, triggering a Th1/Th17 phenotype with the production of proinflammatory cytokines such as TNF-α, INF-γ, interleukin (IL)-18, and IL-21, as well as a B cell response [96].

Strategies to block HLA-DQ2/DQ8 are being studied but remain in the preclinical phase. One such strategy is to deliver competitive inhibitors with gluten peptide analogues that present higher binding affinity than gluten while also not being recognized by T cells. Those inhibitors showed mild blunting of T cell activation in silico [97] and in vitro [98,99]. The quest for such therapy has been challenging, with issues regarding rapid degradation of the peptide ligand and possible interference with other vital MHC immune surveillance functions [100,101].

Another technique for immune modulation is the inhibition of lymphocyte trafficking and homing to the small bowel. This can be achieved by targeting the adhesion molecules on gut endothelial cells (mucosal addressin cellular adhesion molecule-1, MadCAM-1) or their counterpart integrin receptors in lymphocytes (integrin receptor α4β7), as well as by targeting tissue-specific chemokine receptors on lymphocytes (chemokine receptor-9, CCR9) (Table 2).

PTG-100 is an orally administered, potent, and selective α4β7 peptide antagonist. It was already tested in a phase 2a RCT in patients with ulcerative colitis and showed high gastrointestinal exposure, limited systemic exposure, and a dose–response improvement in endoscopy and histology [104]. A phase 1b study (NCT 04524221) evaluating PTG-100 at a dose of 600 mg twice a day versus placebo in 30 CD patients on a gluten challenge was completed in April 2022. Vedolizumab is an anti-α4β7 antibody that is widely used in inflammatory bowel disease. A large epidemiological study found that patients with inflammatory bowel disease treated with steroids, 5-aminosalicylates, and immunomodulators (but not vedolizumab) had a lower risk of developing CD compared to untreated patients, which may suggest a lack of efficacy when using vedolizumab in the treatment of CD. A phase 2 RCT evaluating vedolizumab in CD started in 2016 but was terminated in 2018 due to lack of enrollment (NCT02929316).

Vercirnon is an oral selective antagonist of CCR9 that has shown promising results in early studies of its use in the treatment of Crohn’s disease [105], even though subsequently a phase 3 RCT failed to show efficacy when used for induction therapy [106]. A phase 2 study of vercirnon in CD was completed in 2008 (NCT00540657), but its results were not published.

IL-15 seems to be a pivotal cytokine in the pathogenesis of CD [90]. IL-15 is produced by APCs and epithelial cells and induces activation and proliferation of IEL, thus promoting villous atrophy [102]. The IL-15 receptor consists of three chains: (a) a common cytokine receptor γ-chain that is shared with the receptors for IL-2, IL-4, IL-7, IL-9, and IL-21; a β-chain (IL-15Rβ) that is shared with the IL-2 receptor; and an IL-15 specific α-chain (IL-15Rα) [71]. The IL-15 receptor signals through the Janus kinase–signal transducer and activator of transcription (JAK/STAT) [107].

PRN-015, previously dubbed AMG714, was the first anti-IL-15 evaluated in CD in a proof-of-concept phase 2a RCT [102]. PRN-015 is a fully human IgG1k monoclonal antibody that binds to IL-15. This study evaluated 64 CD patients on a 2–4 g daily gluten challenge who were subcutaneously administered the antibody every other week for 10 weeks. According to the study, the antibody was well tolerated. Compared to placebo, PRN-015 treatment was associated with a decrease in IEL and a symptomatic benefit (less diarrhea development), but failed to show a benefit in terms of serology or Vh:Cd [102]. PRN-015 was also evaluated in 28 patients with type II RCD in a phase 2a RCT, and while it was associated with improvement in diarrhea, it showed no histological benefit over placebo [103]. Importantly, in those patients, PRN-015 was associated with significant adverse events, such as tuberculosis and cerebellar syndrome. An ongoing phase 2b RCT (NCT 04424927) in patients with CD who are non-responsive to a GFD is estimated to be completed in December 2023.

Regarding targeting the IL-15 receptor, a phase 1 RCT of a humanized monoclonal antibody against IL-15Rβ (Hu-Myk-β1) was completed among five patients with RCD in December 2019, but its results have not been published (NCT 01893775).

Tofacitinib is an oral small molecule pan-JAK inhibitor. In preclinical studies with transgenic mice that overexpressed IL-15 and thus developed the pathologic features of CD, tofacitinib was able to revert enteropathy [108]. Case reports also showed histological improvement in a CD patient on a gluten-containing diet [109], as well as in a patient with type II RCD [110]. A phase 2 open-label trial of tofacitinib in type II RCD is ongoing (Eudra CT: 2018-001678-10). (Table 2).

Infliximab, an anti-TNF-α antibody and a cornerstone in the treatment of inflammatory bowel disease, showed encouraging results in case reports of steroid-unresponsive RCD patients [111,112,113,114].

Steroids, particularly budesonide, may have a role in the treatment of non-responsive patients or those with RCD despite the frequent relapse after tapering suggested by case reports and case series [115,116,117]. In newly diagnosed CD however, a pilot study among 27 patients failed to show any benefit from budesonide as an adjunctive treatment to a GFD when attempting to accelerate intestinal mucosal recovery [118].

### 3.5. Inducing Immune Tolerance

CD is characterized by a loss of immune tolerance, with T regulatory cells (Treg) unable to suppress effector T cells [119,120,121]. Four strategies have been studied to re-establish immune gluten tolerance in patients with CD: (a) desensitization through the presentation of gliadin proteins in nanoparticles [122] or (b) red-blood cell moieties [123], (c) therapeutic vaccination [124], or (d) infestation with helminths [125]. These approaches have the advantage of avoiding impairments to systemic immune function [15] (Table 3).

TAK-101, previously dubbed TIMP-GLIA, consists of gliadin encapsulated in negatively charged poly(dl-lactide-co-glycolic acid) nanoparticles. After intravenous administration, TAK-101 is uptaken by APCs in the liver and spleen, which modulates their transcription towards an anti-inflammatory phenotype with (a) downregulation of costimulatory molecules CD80 and CD86, (b) induction of inhibitory PD-L1, and (c) enhanced production of the regulatory cytokines IL-10 and tumor growth factor-β (TGF-β) [122]. In mouse models of CD, TAK-101 induced effector CD4^+^ T cell anergy and CD4^+^ Treg activation, as well as, gliadin-specific unresponsiveness, thus preventing gluten-associated enteropathy [122]. In humans, TAK-101 also downregulates the expression of gut-homing (α4β7^+^ CD4^+^) and gut-retaining (αEβ7^+^ CD8^+^) integrins in circulating T cells [126]. Recently, a phase 2 study evaluating TAK-101 in 33 HLA-DQ2/8 CD patients subjected to a gluten challenge was conducted as a proof-of-concept, consequently showing the induction of antigen-specific immune tolerance in an autoimmune disease. Compared to placebo, infusions of TAK-101 on day 1 and day 8 blunted the growth of a population of gluten-specific INF-γ producing cells by 88%. It also abrogated the flattening of Vh:Cd [126]. An ongoing phase 2 dose-ranging trial (NCT 04530123) among 168 CD patients with a gluten-supplemented diet is estimated to be completed in January 2024.

KAN-101 is still in development and combines erythrocytes and gluten moieties. Red blood cells undergo early apoptosis. Dying erythrocytes combined with gluten are recognized by immune cells, which might induce gluten-specific tolerance [123]. A phase 1 study evaluating the safety and tolerability of KAN-101 is currently ongoing (NCT 04248855).

Nexvax-2 is a therapeutic vaccine that is an adjuvant-free mixture of three peptides (NPL001, NPL002, and NPL003) with immunodominant epitopes for gluten-specific CD4^+^ cells [124] and is used with the aim of rendering those cells unresponsive to further antigen exposure. Phase 1 studies have shown nexvax-2 to be safe and well tolerated after intradermal administration, even though it induced gastrointestinal symptoms such as diarrhea and nausea that are similar to the effects of gluten exposure [127]. However, a subsequent phase 1 RCT with increasing doses, ranging from 60 µg to 150 µg twice weekly for 8 weeks, could not prevent intestinal histological deterioration (assessed by Vh:Cd) in 108 CD patients exposed to a gluten challenge [124]. The intradermal administration may have hit the wrong target, being uptaken by APCs in the skin rather than the spleen and liver [126]. More recently, a phase 2 study with a similar design was terminated prematurely after an interim analysis determined futility [128].

Inoculation with helminths, namely *Necator americanus*, is a strategy that has also been studied in the treatment of inflammatory bowel disease [130]. In celiac disease, two small phase 1 single arm studies showed that *N. americanus* inoculation in CD patients exposed to progressively higher gluten challenges induced a shift in the adaptive T cell response to gluten towards a type 2 phenotype, while also leading to decreased INF-γ and IL-17 response to gluten exposure. Furthermore, compared to historical controls, infestation blunted serological and histological gluten-induced injury [131]. However, infestation failed to improve symptoms [125]. *N. americanus* was well tolerated, except for short-term pruritus at the inoculation site and self-limited abdominal pain [125].

More recently, a phase 2 RCT controlled with placebo that used a similar design among 54 CD patients was disappointing. Infestation did not protect from gluten-induced mucosal deterioration and was associated with a decrease in quality of life scores [129].

## 4. Conclusions

Currently, the only proven effective treatment for CD is a lifelong GFD. Drug development faces many challenges when there is already an established non-pharmacological therapy for a disease. For a new drug to be able to replace a GFD in diet-responsive CD patients, it must be devoid of potential adverse effects, must be simple to administer (preferably as an oral therapy), and must be inexpensive.

Pharmacological treatments for CD may be more useful, at least in the short term, for patients unresponsive to a GFD and may also be useful as an adjunctive treatment in association with a GFD, especially considering the high rate of gluten contamination in the diet due to either inadvertent consumption or non-compliance. To mitigate gluten contamination in the diet, two drugs currently present the most advanced clinical research: larazotide and latiglutenase. Larazotide is a drug that stabilizes enterocyte tight junctions with the aim of decreasing intestinal permeability. While RCTs could not demonstrate a decrease in intestinal permeability due to high variability in the assay, it did decrease symptoms and serology in phase 2 studies, suggesting an effective decrease in the amount of gluten the immune system is exposed to. However, in 2022, a phase 3 trial was suspended after an interim analysis showed no meaningful effect. Latiglutenase is a mix of glutenases that, in phase 2 studies, has been shown to prevent mucosal degradation and symptom development as a result of gluten contamination or challenge. Latiglutenase is a strong contender to become a standard adjunctive therapy in the treatment of CD.

For patients who are unresponsive to a GFD or those with RCD, one key focal point of research is the IL-15 pathway. Blocking IL-15 with a directed monoclonal antibody (PRN15) showed somewhat disappointing results, with the potential for meaningful adverse side effects. More promising seems to be tofacitinib, a pan-JAK inhibitor that acts on the IL-15 receptor signaling pathway. The use of monoclonal antibodies acting on different cytokines and lymphocyte trafficking are still only in embrionary phases of research.

An appealing strategy would be to induce immune tolerance to gluten, and hence avoid the systemic suppression of the immune system. However, two strategies were disappointing: therapeutic vaccines and hookworm infestation. Nevertheless, desensitization to antigen presentation is still in the running.

Lastly, it is still yet to be determined whether the drugs in the pipeline for the treatment of CD may have a role in the treatment of extraintestinal manifestations and conditions associated with CD, such as neuropsychiatric and autoimmune diseases.

## Figures and Tables

**Table 1 ijms-24-00945-t001:** Relevant clinical trials on agents that block immunogenic gluten exposure.

Agent	Study	Trial Phase	Population	Treatment	Duration	Main Results (vs. Placebo)
*Endopeptidases*						
Latiglutenase	Tye-Din, 2010 [59]	1	20 CD patients on a gluten challenge (16 g/day)	800 mg/day vs. placebo	3 days	↓ INF-γ secretion by gluten-specific T cells in peripheral blood
	Lahdeaho, 2014 [60]	2a	41 CD patients on a gluten challenge (2 g/day)	900 mg/day vs. placebo	6 weeks	Prevented mucosal deterioration (no ↓ Vh:Cd or ↑ IEL) No improvement in symptoms
	Murray, 2017 [61]; Syage, 2017 [62]	2b	494 CD patients with moderate or severe symptoms on a GFD ≥ 1 year	100 mg, 300 mg, 450 mg, 600 mg, or 900 mg/day vs. placebo	12 or 24 weeks	No ≠ in Vh:Cd or ↑ IEL No ≠ in serology Improvement in the symptoms of seropositive patients with ≥600 mg/day
	Murray, 2022 [63]	2b	43 CD patients on a gluten challenge (2 g/day)	1200 mg/day vs. placebo	6 weeks	Prevented mucosal deterioration (lower ↓ Vh:Cd) Tendency to decrease symptoms
	NCT 04243551	2b	120 symptomatic CD patients undergoing periodic gluten exposure	vs. placebo	6 weeks	Ongoing Estimated completion May 2023
AN-PEP	Tack, 2013 [64]	2	14 CD patients on a gluten challenge (7 g/day)	Topping with either AN-PEP or placebo	2 weeks	No ≠ in Vh:Cd or ↑ IEL No ≠ in terms of quality of life
	NCT 04788797	4	14 CD patients on a daily gluten challenge	2 capsules/day vs. placebo	8 weeks	Ongoing Completion December 2022
TAK-062	Pultz, 2021 [65]	1	CD in GFD and healthy subjects after a gluten meal (3–9 g)	100–900 mg	6 weeks	Well tolerated
*Gluten sequestration*						
AGY	Sample, 2017 [66]	1	10 CD patients on a GFD	1000 mg bid	4 weeks	↓ symptoms ↓ serology ↓ LMER
	NCT 03707730	2	Symptomatic CD patients on a GFD	Before meals vs. placebo	14 weeks	Ongoing Completion December 2022
BL-7010	NCT 01990885	1	40 asymptomatic CD patients	Dose-finding	Single dose	Completed in 2014 No data published so far
*Tight junction modulation*						
Larazotide acetate	Paterson, 2007 [67]	1	21 CD patients on a 1-day gluten challenge (2.5 g)	12 mg vs. placebo	3 days	↓ INF-γ secretion ↓ symptoms No ≠ in LMAR
	Leffler, 2012 [68]	2a	86 CD patients ± gluten challenge (2.4 g/day)	0.25 mg, 1 mg, 4 mg, or 8 mg/day vs. placebo	14 days	↓ symptoms No ≠ in LMAR
	Kelly, 2013 [69]	2b	177 CD patients on a gluten challenge (2.7 g/day)	1 mg, 4 mg, or 8 mg/day vs. placebo	6 weeks	↓ symptoms ↓ serology No ≠ in LMER
	Leffler, 2015 [70]	2b	342 symptomatic CD patients on a GFD	0.5 mg, 1 mg, or 2 mg/day vs. placebo	12 weeks	↓ symptoms
	NCT 03569007	3	307 symptomatic CD patients on a GFD	0.5 mg or 1 mg vs. placebo	12 weeks	Prematurely interrupted after interim analysis in June 2022

**Table 2 ijms-24-00945-t002:** Relevant clinical trials on agents that induce immune modulation.

Agent	Study	Trial Phase	Population	Treatment	Duration	Main Results (vs. Placebo)
*Lymphocyte trafficking*						
PTG-100 (anti-α4β7)	NCT 04524221	1b	30 CD patients on a gluten challenge	600 mg bid vs. placebo	42 days	Completed in April 2022 No data published so far
Vedolizumab (anti-α4β7)	NCT 02929316	2	CD patients on a gluten challenge	300 mg vs. placebo	6 weeks	Terminated in 2018 due to lack of enrollment
Vercinon (anti-CCR9)	NCT 00540657	2	30 CD patients on a gluten challenge	250 mg bid vs. placebo	13 weeks	Completed in 2008 No data published so far
*IL-15 targeting*						
PRN-015 or AMG714 (anti-IL-15)	Lähdeaho, 2019 [102]	2a	64 CD patients on a gluten challenge (2–4 g/day)	150 mg, 300 mg/day vs. placebo	12 weeks	↓ symptoms (diarrhea) ↓ IEL at 300 mg No ≠ in serology or Vh:Cd
	Cellier, 2019 [103]	2a	Type II RCD	8 mg/kg 2×/week vs. placebo	12 weeks	↓ symptoms (diarrhea) No ≠ in IEL, aberrant IEL, or Vh:Cd Adverse events: 26% vs. 11%
	NCT 04424927	2b	220 CD patients non-responsive to a GFD	3 ≠ arms vs. placebo	28 weeks	Ongoing Completion December 2023
Hu-Mik-β1 (anti-IL15Rβ1)	NCT 01893775	1	5 RCD patients	Every 3 weeks	9 weeks	Completed in December 2019 No data published so far
Tofecitinib (pan-JAK inhibitor)	Eudra CT: 2018-001678-10	2	Type II RCD (open-label) patients	10 mg bid	12 weeks	Ongoing

**Table 3 ijms-24-00945-t003:** Relevant clinical trials on agents that induce immune tolerance.

Agent	Study	Trial Phase	Population	Treatment	Duration	Main Results (vs. Placebo)
*Nanoparticles for gliadin presentation*						
TAK-101 (TIMP-GLIA)	Kelly, 2021 [126]	2	33 CD patients on a gluten challenge	8 mg day 1 and 8 vs. placebo	5 weeks	↓ growth of gluten-specific INF-γ producing cells ↓ Vh:Cd flattening
	NCT 04530123	2	168 CD patients on a gluten challenge	1–4 mg day 1 and 8 vs. placebo	20 weeks	Ongoing Completion January 2024
*Gluten erythrocyte moiety on red blood cells*						
KAN-101	NCT 04248855	2b	41 CD patients	Dose-ranging vs. placebo	4 weeks	Completed in December 2021 No data published so far
*Therapeutic vaccine*						
Nexvax-2	Daveson, 2017 [127]	1	36 HLA-DQ2.5 CD patients on a GFD	3–900 µg 2×/week ID vs. placebo	6 weeks	Transient symptoms resembling a gluten challenge Safe and well tolerated
	Goel, 2017 [124]	2	108 HLA-DQ2.5 CD patients on a gluten challenge	60–150 µg 2×/week ID vs. placebo	8 weeks	Transient symptoms resembling a gluten challenge No ≠↓ Vh:Cd
	Truitt, 2019 [128]	2	146 HLA-DQ2.5 CD patients on a gluten challenge	32 doses SC vs. placebo	26 weeks	Terminated 2019 for futility
*Helminth infestation*						
*Necator americanus*	Daveson, 2011 [125]	1	10 CD patients on a gluten challenge (16 g/day)	vs. historical controls	21 weeks	Short-term pruritus at the inoculation site No improvements in symptoms or histology
	Croese, 2020 [129]	2	54 CD patients subjected to gluten intake	vs. placebo	42 weeks	No histology improvement ↓ quality of life

## Data Availability

Not applicable.

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
