# Peer review of "New Developments in Celiac Disease Treatment"

_ijms, 2023, doi:10.3390/ijms24020945_

Round 1

Reviewer 1 Report

In my opinion, this is an interesting review with clinical impact discussing the explored treatment approach for celiac disease. 

I have minor point to suggest.

-Regarding Larazotide, the authors can now add that the Phase III trial studying larazotide for the treatment of celiac disease has been discontinued last June due to disappointing interim results. 

-The authors should also discuss the impact of treatments on the associated disorders that not rarely celiac patients can present. As previous studies also reported potential improvement of associated disorders after starting gluten free-diet, new adjunctive treatments should also be evaluated for the impact on associated disorders. In particular, autoimmune disorders such as diabetes, thyroiditis (Coeliac disease in patients with autoimmune thyroiditis. Digestion. 2001;64(1):61-5.), autoimmune liver diseases (Celiac disease in autoimmune cholestatic liver disorders. Am J Gastroenterol. 2002 Oct;97(10):2609-13.), hypertransaminasemia of unknown origin (Anti tissue transglutaminase antibodies as predictors of silent coeliac disease in patients with hypertransaminasaemia of unknown origin. Dig Liver Dis. 2001 Jun-Jul;33(5):420-5.), atopy (Prevalence of silent coeliac disease in atopics. Dig Liver Dis. 2000 Dec;32(9):775-9.), neurological disorders and neurological-related autoantibodies (Anti-ganglioside antibodies and celiac disease. Allergy Asthma Clin Immunol. 2021 May 28;17(1):53). It would be clinically relevant to verify the impact of new treatments on such a category of patients. 

Author Response

I thank the editor and the reviewers for the constructive critiques and for the opportunity to improve the manuscript. We hope the reviewers now find it worthy for publication.

It follows the point-by-point response to the reviewers’ comments.

Reviewer #1

In my opinion, this is an interesting review with clinical impact discussing the explored treatment approach for celiac disease. I have minor point to suggest.

R: Thank you for the encouraging words.

  1. Regarding Larazotide, the authors can now add that thePhase III trial studying larazotide for the treatment of celiac diseasehas been discontinued last June due to disappointing interim results. 

R: I thank the reviewer for this important comment. Accordingly, the following was added:

“A phase 3 RCT on symptomatic CD patients on GFD (NCT 03569007), named CeDLara (Celiac Disease Larazotide) planned to enroll 525 patients. However, an interim analysis of half of the initial target enrollment showed disappointing results, leading to discontinuation of the study in June 2022”

  1. The authors should also discuss the impact of treatments on the associated disorders that not rarely celiac patients can present. As previous studies also reported potential improvement of associated disorders after starting gluten free-diet, new adjunctive treatments should also be evaluated for the impact on associated disorders.In particular, autoimmune disorders such as diabetes, thyroiditis (Coeliac disease in patients with autoimmune thyroiditis. Digestion. 2001;64(1):61-5.), autoimmune liver diseases (Celiac disease in autoimmune cholestatic liver disorders. Am J Gastroenterol. 2002 Oct;97(10):2609-13.), hypertransaminasemia of unknown origin (Anti tissue transglutaminase antibodies as predictors of silent coeliac disease in patients with hypertransaminasaemia of unknown origin. Dig Liver Dis. 2001 Jun-Jul;33(5):420-5.), atopy (Prevalence of silent coeliac disease in atopics. Dig Liver Dis. 2000 Dec;32(9):775-9.), neurological disorders and neurological-related autoantibodies (Anti-ganglioside antibodies and celiac disease. Allergy Asthma Clin Immunol. 2021 May 28;17(1):53). It would be clinically relevant to verify the impact of new treatments on such a category of patients. 

R: I thank the reviewer for this insightful comment. Information regarding the effect of gluten free diet was added as following: “CD associates with a panoply of extraintestinal manifestations and autoimmune diseases such as neuropsyquiatric20-22 and dermatological manifestations23, unexplained abnormal liver enzymes24, type 1 diabetes mellitus and autoimmune thyroid disorders25. Adherence to GFD may improve most of those manifestations. Regarding neuropsychiatric manifestations, it is known to improve headache26, and gluten ataxia as long as there is no irreversible loss of Purkinje cells in the cerebellum27. GFD may also improve response to drug-resistant epilepsy28, gluten-induced cognitive impairment29, psychiatric disorders particularly anxiety26, and fatigue in up to 50% the patients30, while response to peripheral neuropathy is variable with lower response rates in patients presenting anti-neuronal antibodies31. Dermatitis herpetiformis typically responds to GFD32. Psoriasis33, and chronic urticaria34, recurrent aphthous stomatitis may also improve in patients with CD, while dental enamel defects are irreversible35. Furthermore, GFD may improve metabolic control in CD patients with type 1 diabetes mellitus33, bone density36, menstrual disturbances and fertility37 and unexplained transaminitis38.”

Regarding the effect of new treatments on extraintestinal manifestations or associated conditions is yet to be studied. This was added to the conclusion: “Lastly, it is yet to be determined if the drugs in the pipeline for the treatment of CD may have a role in the treatment of extraintestinal manifestations and conditions associated with CD such as neuropsychiatric and autoimmune diseases.”

Reviewer 2 Report

In this review, the authors addressed the rationale that has been applied for celiac disease (CD) drug development and the clinical advances in its research. They discuss gluten-free diet as the standard of care, non-immunogenic gluten delivery, strategies blocking immunogenic gluten exposure and inhibiting transglutaminase and immune modulation also by inducing immune tolerance.

The topic is of current clinical interest and it is well organized. However, in a review addressing the novelty of celiac disease and addressing the potential impact, some topics should be recalled and discussed.

-Gluten-free diet: the authors should highlight that GFD is currently considered the only proven treatment for CD as recommended by current international guidelines, as recently well summarized (Current guidelines for the management of celiac disease: A systematic review with comparative analysis. World J Gastroenterol. 2022 Jan 7;28(1):154-175).

-The authors properly stated that one to two-thirds of adults will not achieve histological recovery after one year on strict GFD. In this regard, it would be of clinical relevance to recall the importance of serological markers, of mucosal damage, such as anti-actin IgA antibodies, that may be useful to better follow-up CD patients after starting GFD, as previously demonstrated (Anti-actin IgA antibodies in severe coeliac disease. Clin Exp Immunol. 2004 Aug;137(2):386-92).

-Transglutaminase inhibition: the authors should also underline that such a new treatment strategy has been proposed as adjunctive treatment (together with standard GFD) in CD patients who exhibit persistent symptoms also under GFD.

-The last point deserving to be recalled is the impact of GFD and new treatments on the so-called secondary autoimmunity in CD patients. In this regard, the authors should recall the high prevalence of other autoimmune serological markers as well as autoimmune diseases that may benefit from GFD such as neurological disorders, hypertransaminasemia and other as previously demonstrated (Sera of patients with celiac disease and neurologic disorders evoke a mitochondrial-dependent apoptosis in vitro. Gastroenterology. 2007 Jul;133(1):195-206; Anti-ganglioside antibodies in coeliac disease with neurological disorders. Dig Liver Dis. 2006 Mar;38(3):183-7; Anti tissue transglutaminase antibodies as predictors of silent coeliac disease in patients with hypertransaminasaemia of unknown origin. Dig Liver Dis. 2001 Jun-Jul;33(5):420-5.).

Author Response

I thank the editor and the reviewers for the constructive critiques and for the opportunity to improve the manuscript. We hope the reviewers now find it worthy for publication.

It follows the point-by-point response to the reviewers’ comments.

Reviewer #2

In this review, the authors addressed the rationale that has been applied for celiac disease (CD) drug development and the clinical advances in its research. They discuss gluten-free diet as the standard of care, non-immunogenic gluten delivery, strategies blocking immunogenic gluten exposure and inhibiting transglutaminase and immune modulation also by inducing immune tolerance. The topic is of current clinical interest and it is well organized.

R: Thank you for the encouraging words.

However, in a review addressing the novelty of celiac disease and addressing the potential impact, some topics should be recalled and discussed.

  1. Gluten-free diet: the authors should highlight that GFD is currently considered the only proven treatment for CD as recommended by current international guidelines, as recently well summarized (Current guidelines for the management of celiac disease: A systematic review with comparative analysis. World J Gastroenterol. 2022 Jan 7;28(1):154-175).

R: The first sentence in the conclusions is precisely: “Currently, the only proven effective treatment for CD is lifelong GFD”. We highlighted this point in the chapter of GFD, after the reference the reviewer cited: “Patients with CD independently of the presence of symptoms, symptomatic potential CD (that is patients with positive anti-transglutaminase antibody and normal duodenal histology), dermatitis herpetiformis, and gluten ataxia should follow a lifelong GFD18, since GFD is currently the only proven treatment for CD.”

  1. The authors properly stated that one to two-thirds of adults will not achieve histological recovery after one year on strict GFD. In this regard, it would be of clinical relevance to recall the importance of serological markers, of mucosal damage, such as anti-actin IgA antibodies,that may be useful to better follow-up CD patients after starting GFD, as previously demonstrated (Anti-actin IgA antibodies in severe coeliac disease. Clin Exp Immunol. 2004 Aug;137(2):386-92).

R: It was added: “Of note, monitoring of anti-actin IgA antibodies may help to predict GFD-induced resolution of intestinal mucosa damage51.”

  1. Transglutaminase inhibition: the authors should also underline that such a new treatment strategy has been proposed as adjunctive treatment (together with standard GFD) in CD patients who exhibit persistent symptoms also under GFD.

R: I am not aware of clinical trials published or ongoing on transglutaminase-2 inhibitors in CD patients on GFD not submitted to a gluten challenge. Could the reviewer specify?

  1. The last point deserving to be recalled is the impact of GFD and new treatments on the so-called secondary autoimmunity in CD patients. In this regard, the authors should recall the high prevalence of other autoimmune serological markers as well as autoimmune diseases that may benefit from GFD such as neurological disorders, hypertransaminasemia and other as previously demonstrated (Sera of patients with celiac disease and neurologic disorders evoke a mitochondrial-dependent apoptosis in vitro. Gastroenterology. 2007 Jul;133(1):195-206; Anti-ganglioside antibodies in coeliac disease with neurological disorders. Dig Liver Dis. 2006 Mar;38(3):183-7; Anti tissue transglutaminase antibodies as predictors of silent coeliac disease in patients with hypertransaminasaemia of unknown origin. Dig Liver Dis. 2001 Jun-Jul;33(5):420-5.).

R: I thank the reviewer for this insightful comment. Information regarding the effect of gluten free diet was added as following: “CD associates with a panoply of extraintestinal manifestations and autoimmune diseases such as neuropsyquiatric20-22 and dermatological manifestations23, unexplained abnormal liver enzymes24, type 1 diabetes mellitus and autoimmune thyroid disorders25. Adherence to GFD may improve most of those manifestations. Regarding neuropsychiatric manifestations, it is known to improve headache26, and gluten ataxia as long as there is no irreversible loss of Purkinje cells in the cerebellum27. GFD may also improve response to drug-resistant epilepsy28, gluten-induced cognitive impairment29, psychiatric disorders particularly anxiety26, and fatigue in up to 50% the patients30, while response to peripheral neuropathy is variable with lower response rates in patients presenting anti-neuronal antibodies31. Dermatitis herpetiformis typically responds to GFD32. Psoriasis33, and chronic urticaria34, recurrent aphthous stomatitis may also improve in patients with CD, while dental enamel defects are irreversible35. Furthermore, GFD may improve metabolic control in CD patients with type 1 diabetes mellitus33, bone density36, menstrual disturbances and fertility37 and unexplained transaminitis38.”

Regarding the effect of new treatments on extraintestinal manifestations or associated conditions is yet to be studied. This was added to the conclusion: “Lastly, it is yet to be determined if the drugs in the pipeline for the treatment of CD may have a role in the treatment of extraintestinal manifestations and conditions associated with CD such as neuropsychiatric and autoimmune diseases.”